# Pasteurisation temperatures effectively inactivate influenza A viruses in milk

Jenna Schafers [1,6], Caroline J. Warren[2,6], Jiayun Yang[3,6], Junsen Zhang [4,6], Sarah J. Cole[4], Jayne Cooper[2], Karolina Drewek[2], B. Reddy Kolli[3], Natalie McGinn[2], Mehnaz Qureshi[3], Scott M. Reid[2], Thomas P. Peacock[3], Ian Brown [3], Joe James [2,5], Ashley C. Banyard [2,5], Munir Iqbal [3], Paul Digard [1] & Edward Hutchinson [4] ✉

In late 2023 an H5N1 lineage of high pathogenicity avian influenza virus (HPAIV) began circulating in American dairy cattle Concerningly, high titres of virus were detected in cows' milk, raising the concern that milk could be a route of human infection. Cows' milk is typically pasteurised to render it safe for human consumption, but the effectiveness of pasteurisation on influenza viruses in milk was uncertain. To assess this, here we evaluate heat inactivation in milk for a panel of different influenza viruses. This includes human and avian influenza A viruses (IAVs), an influenza D virus that naturally infects cattle, and recombinant IAVs carrying contemporary avian or bovine H5N1 glycoproteins. At pasteurisation temperatures of 63 °C and 72 °C, we find that viral infectivity is rapidly lost and becomes undetectable before the times recommended for pasteurisation (30 minutes and 15 seconds, respectively). We then show that an H5N1 HPAIV in milk is effectively inactivated by a comparable treatment, even though its genetic material remains detectable. We conclude that pasteurisation conditions should effectively inactivate H5N1 HPAIV in cows' milk, but that unpasteurised milk could carry infectious influenza viruses.

Since 2020 an H5N1 clade 2.3.4.4b lineage of high pathogenicity influenza virus (HPAIV) has spread rapidly around the world, causing the worst outbreak of avian influenza on record[1–3]. As H5N1 IAVs can cause severe disease in humans[4], the pandemic potential of this outbreak is of great concern[5]. While HPAIVs are able to cross between host species, viral adaptation to sustained transmission within mammal populations is uncommon. The current H5N1 virus has caused repeated spillover infections in mammals, but most of these were in wild animals and not in close proximity to humans[6–8]. This changed in early 2024 when it was realised that H5N1 HPAIVs were spreading among dairy cattle in the USA[9]. This was alarming because of the extensive human-animal interface of the dairy industry, including the widespread consumption of dairy products. It was also surprising, for two reasons. Firstly, cattle had previously been considered resistant to IAV infection, with only sporadic cases reported[10,11]. Secondly, although IAV typically spreads by respiratory or faecal-oral transmission, H5N1 HPAIV was shed at startlingly high titres into milk[12]. Shedding into milk appears to have led to further spillover events on dairy farms, with H5N1 identified in dead farm cats, wild raccoons and foxes, cattle-associated perching birds, and nearby poultry flocks. Furthermore, HPAIV in cows has also resulted in multiple infections of dairy farm workers[3,13,14]. This new route of transmission has also resulted in H5N1 HPAIV being shed into milk sold for human consumption, with viral genetic material detected in as much as 20% of supermarket milk

[1]Roslin Institute, The University of Edinburgh, Easter Bush Campus, Midlothian, UK. [2]Department of Virology, Animal and Plant Health Agency-Weybridge, Woodham Lane, New Haw, Addlestone, Surrey, UK. [3]The Pirbright Institute, Ash Road, Woking, Surrey, UK. [4]MRC-University of Glasgow Centre for Virus Research, Glasgow, UK. [5]WOAH/FAO Reference Laboratory for Avian Influenza, Animal and Plant Health Agency-Weybridge, Woodham Lane, New Haw, Addlestone, Surrey, UK. [6]These authors contributed equally: Jenna Schafers, Caroline J. Warren, Jiayun Yang, Junsen Zhang. ✉e-mail: Edward.Hutchinson@glasgow.ac.uk

in some affected areas, as well as in other milk products such as cheese and ice cream[15]. In response to this, determining if humans could be exposed to infectious H5N1 HPAIV through consuming cows' milk was a matter of urgent importance.

Because cows' milk can carry a variety of pathogens, it is typically pasteurised before human consumption, as well as being homogenised to stabilise the emulsified fats and prevent the milk from separating[16]. Pasteurisation is a well-established method of heat inactivation, which was first formalised by Pasteur for wine in 1864[17] and correlated with drastic falls in infant mortality and other diseases when widely applied to milk over the first half of the twentieth century[18,19]. It was assumed that pasteurisation of milk would also be effective against bovine H5N1 HPAIV, but this was based on general assumptions about the structure of the virus and on the very limited prior studies of heat treatment of other influenza viruses suspended in other substances[20–23]. Encouragingly, initial reports indicated that infectious influenza virus could not be recovered from pasteurised milk containing viral genetic material[3,15,24], but without a general understanding of how influenza viruses in milk respond to pasteurisation, it was hard to predict the robustness of commercial pasteurisation against this new strain of virus.

Here, we answer this question by determining the general response of influenza viruses to pasteurisation times and temperatures in milk. We compare our results to other studies carried out in parallel and conclude that pasteurisation is likely to be highly effective at inactivating influenza viruses in milk. As the consumption of unhomogenised and unpasteurised (raw) milk is also popular in some affected areas, we also assessed whether influenza viruses remain infectious in milk if heating is not applied, showing that raw milk is capable of carrying infectious influenza viruses.

## Results

To assess the effects of pasteurising temperatures on influenza viruses, we first tested the responses of a variety of influenza virus strains (Table 1)[25] at biosafety containment level 2. We also used reverse genetics to generate a panel of 6:2 reassortant influenza viruses carrying the internal genes of the laboratory strain A/Puerto Rico/8/1934 (PR8) and the surface proteins (HA and NA) of various representatives of H5N1 clade 2.3.4.4b, all de-engineered to replace the polybasic

cleavage site that renders them highly pathogenic with a monobasic cleavage site (Table 2)[26]. For all viruses, to mimic the effects of commercial pasteurisation we applied pasteurising temperatures for specific time intervals by mixing the virus 1:10 with milk and heating small volumes in thin-walled PCR tubes in a thermocycler. The milk was then rapidly cooled, diluted in tissue culture medium and infectivity was assessed by plaque assay. Our aim was not to test specific models of pasteurisation equipment, but rather to determine how quickly inactivation of influenza viruses occurred at the temperatures required for a well-conducted pasteurisation.

We chose temperatures representing the two most common methods of pasteurising milk: low-temperature long time (LTLT; the vat method), which requires heating to at least 62.5 °C (in our study, 63 °C) for at least 30 min[27]; and high-temperature short time (HTST), which requires heating to at least 72 °C for at least 15 s[28].

We first tested PR8 and an H5N3 avian influenza virus in both raw milk and commercially available pasteurised, homogenised whole milk (processed milk). We observed similar, rapid inactivation on heating in both cases (Fig. 1a). This effect was comparable when spiking virus into fresh milk that had been stored at 4 °C and thawed milk that had been stored at −20 °C (Fig. 2). We therefore tested our remaining panel of viruses in processed milk.

We tested the panel of viruses using a range of heating times to assess the rates of inactivation at 63 °C and at 72 °C (Fig. 1b). At both 63 °C and 72 °C the infectivity of all viruses was rapidly lost, dropping by orders of magnitude in seconds. While there were some differences between the viruses tested, in all cases, infectivity fell below the limit of detection well in advance of the minimum times required for milk pasteurisation.

We then used PR8 to test pasteurisation in different types of shop-brought milk, as we were concerned that different cream concentrations obtained by processing milk might alter the effectiveness of the process[29]. At each temperature we observed similar kinetics of inactivation regardless of whether we mixed the virus with skimmed (0.1% w/v fat), semi-skimmed (1.7% w/v fat) or whole (3.6% w/v fat) milk (Fig. 1c). Although adding virus back to processed milk may not fully capture the complexity of separating milk and cream in dairy production[16], these results suggest that our findings are likely to be robust across different milk types.

### Table 1 | Influenza viruses used in the study

| Strain name | Short Name | Details |
|---|---|---|
| A/Puerto Rico/8/1934 (H1N1) | PR8 (PR8:PB) | Laboratory strain (PR8 refers to data collected at the Roslin Institute, and PR8:PB to data collected at the Pirbright Institute) |
| BrightFlu | BF-PR8 | A PR8 derivative encoding a fluorescent marker (data collected at the MRC-University of Glasgow Centre for Virus Research)[25] |
| A/wild-duck/Italy/17VIR6926-1/2017 H5N2 (H5N2) | H5N2 | low pathogenicity avian influenza virus |
| A/Duck/Singapore/97 (H5N3) | H5N3 | low pathogenicity avian influenza virus[58] |
| D/bovine/France/5920/2014 | IDV | a separate genus of influenza virus that naturally infects cattle |
| A/chicken/Scotland/054477/2021 | H5N1 | high pathogenicity avian influenza virus |

### Table 2 | Reassortant influenza viruses uses in the study

| Source of HA and NA | Short Name | Details of HA and NA |
|---|---|---|
| A/chicken/Scotland/054477/2021 | PR8:AIV09 | AIV09 (AB genotype) |
| A/chicken/England/085598/2022 | PR8:AIV48 | AIV48 (BB genotype) |
| A/dairy cow/Texas/24-008749-001-original/2024 | PR8:Cattle | cattle isolate |
| A/goat/Minnesota/24-007234-003-original/2024 | PR8:Goat | goat isolate |

Reassortant viruses contain an NA and a de-engineered HA from the strain indicated, with the remaining genes from the laboratory strain PR8.

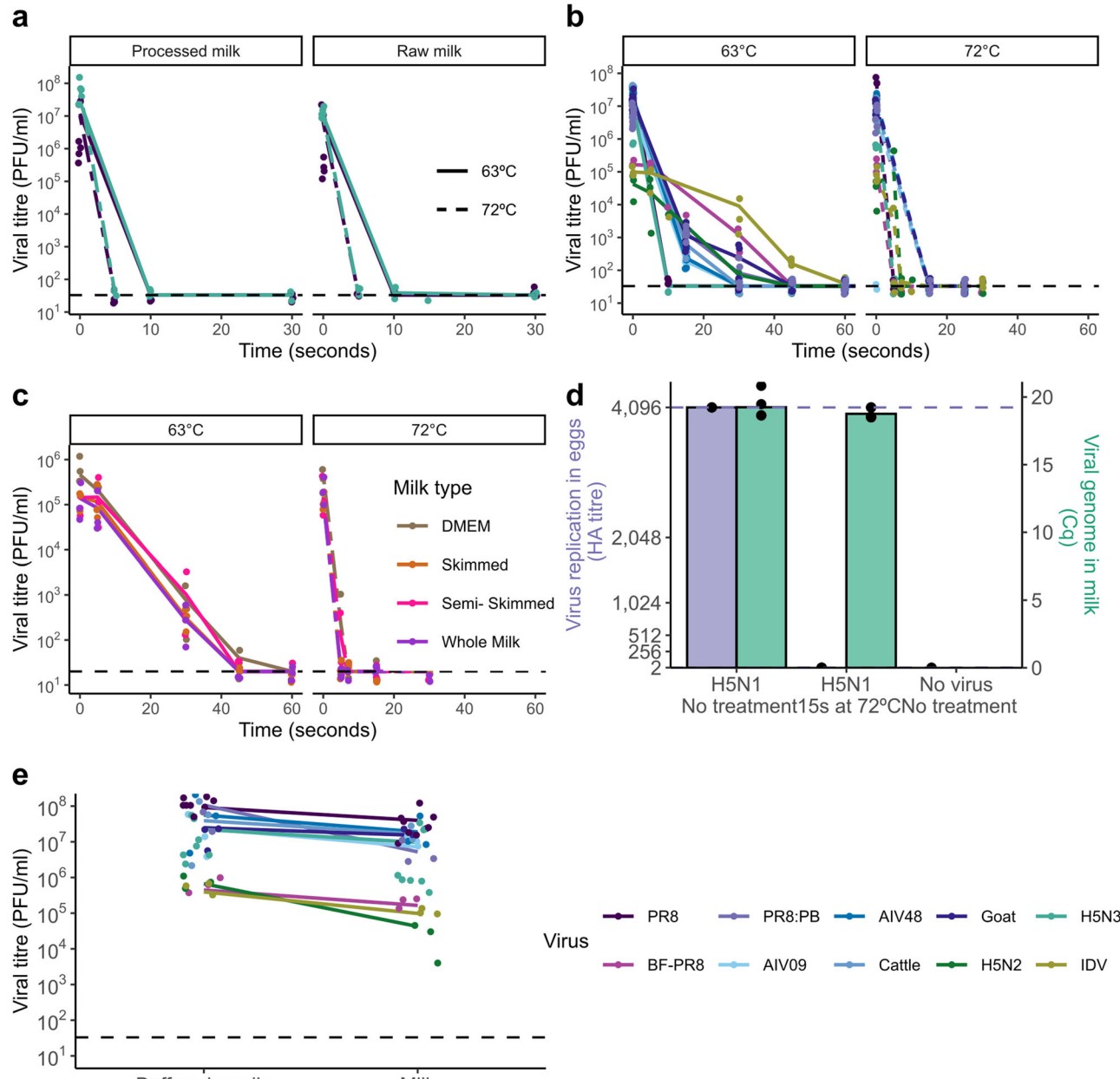

**Fig. 1 | Pasteurisation effectively inactivates influenza viruses in milk. a** PR8 and H5N3 were mixed with raw milk or shop-bought pasteurised whole milk (processed milk), heated for the indicated time and then cooled. Infectivity was measured by plaque assay. Three independent repeats are shown, plotting the mean of duplicate measurements; lines connect the mean value for each condition. Limit of detection (LoD) = 33 PFU/ml. **b** Viruses were mixed with processed milk and treated as in (**a**). Three independent repeats are shown; lines connect the mean value for each condition. For BF-PR8, H5N2 and IDV LoD = 20 PFU/ml, for PR8, PR8 reassortants and H5N3 LoD = 33 PFU/ml. **c** The virus BF-PR8 was mixed with processed milk of differing fat concentrations, or with tissue culture medium, and then treated as in (**a**). Three independent repeats are shown; lines connect the mean value for each condition. LoD = 20 PFU/ml (**d**) H5N1 HPAIV was mixed with raw milk, either unheated or pre-heated to 71.7 °C, then cooled after 15 s and used to inoculate three replicate eggs. Viral replication in eggs was assessed by haemagglutination assay (upper and lower LoD are $2^{12}$ and $2^1$ HAU, respectively). Viral genome in milk was detected using the H5 HPAIV rRT-PCR assay. For each of three independent repeats the individual Cq values of the milk and the mean HA titres of three replicate eggs are shown, along with bars showing the mean values of these measurements. **e** Comparison of the plaque titres of influenza viruses when mixed with tissue culture medium/phosphate-buffered saline, or with milk. Data are shown for 7 (H5N3), 8 (PR8) or 3 (all other viruses) independent repeats. Details of viruses are given in Tables 1 and 2. Source data are available file at https://osf.io/m4fa5/.

Our experiments in thermocyclers showed that all influenza viruses responded similarly to heating, including IDV, which has been reported to be unusually thermally stable[30]. This strongly suggested that an H5N1 HPAIV in milk would also be inactivated by pasteurisation times and temperatures. We next tested this hypothesis directly, adopting a different experimental design that could be used in a high-containment laboratory.

To do this, we used the wild-type H5N1 strain A/chicken/Scotland/054477/2021 (AIV09/AB genotype) and mimicked the conditions of HTST pasteurisation at SAPO containment level 4. In this experiment, we took raw milk rather than processed milk, either left this unheated or pre-heated it to 71.7 °C and then mixed the milk with one part in 100 of virus (a final titre of $3 \times 10^7$ EID$_{50}$). After 15 s the mixture was cooled on ice, after which viral genomes were detected by RT-PCR and

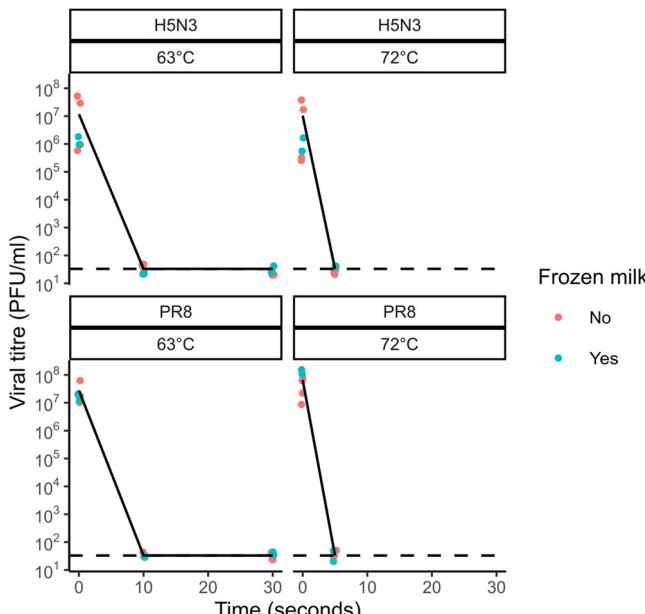

**Fig. 2 | Comparison of fresh and frozen milk.** H5N3 and PR8 viruses were mixed with pasteurised whole milk which had either been stored at 4 °C until use or previously frozen and thawed. Mixtures were then exposed to pasteurisation temperatures for the indicated time. Datapoints indicate the results of 3 (for frozen milk) or 2 (for non-frozen milk) independent repeats, and the overlaid black line indicates the mean decay for both milks. Source data are available at https://osf.io/m4fa5/.

infectivity was assessed by inoculation of milk into embryonated fowls' eggs (EFEs), followed by incubation and a haemagglutination assay of the allantoic fluid (Fig. 1d). Heat treatment did not affect the detection of viral genomes in milk. However, although infectious virus was isolated from room-temperature milk, no infectious virus could be isolated in EFEs following exposure to HTST pasteurisation conditions, either during direct inoculation (Fig. 1d) or when inoculated material was passaged to a second EFE.

Overall, we concluded that heating to pasteurisation temperatures effectively inactivates influenza A and D viruses, including H5N1 HPAIVs, within the times required for pasteurisation.

Finally, we addressed the question of whether raw milk can carry infectious influenza virus. We found that mixing influenza viruses with unheated milk caused a slight reduction in infectivity, consistent with previous studies[11], through a currently unknown mechanism (Fig. 1e). However, it is important to note that this reduction was never more than a slight effect, which is not at all comparable with the orders-of-magnitude reductions in infectivity caused by pasteurisation. Experimental studies have clearly shown that the high titres of H5N1 HPAIV shed into unpasteurised milk can readily deliver an infectious dose of virus by to other animals by oral inoculation[31]. For all viruses tested in the current study, including H5N1 HPAIV and PR8 with H5N1 surface proteins, unpasteurised milk was clearly an effective carrier of infectious influenza viruses (Fig. 1b–d).

## Discussion

In this study, we responded to reports that H5N1 HPAIV had been detected in milk from infected dairy cattle in the USA by asking if pasteurisation of cows' milk could inactivate influenza viruses. Given the urgency of this question, we made two decisions in designing our study which should be considered when interpreting our results. Firstly, rather than assess specific pieces of commercial pasteurisation equipment (as described in ref. 16), we made a general assessment of the times needed to inactivate influenza viruses, by heating in milk at pasteurising temperatures under well-controlled laboratory

conditions. This allowed us to establish general principles which can be used for quality control assessments of specific industrial pasteurisation apparatuses. Secondly, as well as testing the effects of pasteurisation on a recent H5N1 HPAIV, we considered a panel of influenza viruses, including an influenza D virus with a potentially higher thermal tolerance[30]. This allowed us to establish that these conditions should be generally applicable for the inactivation of any influenza virus in milk by pasteurisation.

Overall, we found that pasteurisation temperatures of both 63 °C (LTLT) and 72 °C (HTST) rapidly and effectively inactivated influenza viruses in milk (Fig. 1b–d). In the case of H5N1 HPAIV, treatment at 72 °C eliminated infectivity without affecting the detection of viral genetic material, consistent with reports from the USA that have to date detected viral genetic material but no infectious virus in pasteurised milk[12,32] (Fig. 1d). While it is plausible that homogenisation of milk may also inactivate influenza virus particles, we did not test this in the current study as pasteurisation by itself proved to be extremely efficient at reducing the infectivity of influenza viruses.

During the preparation and revision of this study, a number of other manuscripts were published exploring the effects of pasteurisation on influenza viruses in milk (Table 3[29,33–37]). It is useful to compare all of these studies when making informed decisions about the effectiveness of pasteurisation for inactivating influenza viruses in milk, and we provide a brief summary here to aid this.

In every case, it was found that pasteurisation temperatures rapidly reduced the infectivity of influenza viruses. However, it is clear that the effects of pasteurisation are not instantaneous, and the point at which infectivity became undetectable varied somewhat between studies. It was consistently shown that heating to 63 °C (LTLT method) fully inactivated influenza viruses long before reaching the minimum pasteurisation time of 30 min (Fig. 1a–c and Table 3). Heating to 72 °C (HTST method) also consistently caused very rapid inactivation of the virus, but the times needed for virus titres to drop to the limit of detection were close to the minimum recommended inactivation time of 15 s (Fig. 1a–c). As a result, while several studies, including our own[34,35,37], found that all detectable virus was inactivated by heating for pasteurising times and temperatures, other studies reported low but detectable levels of residual infectivity after heating for times very close to the minimum required for pasteurisation[29,33,36,38,39]. Several factors could account for this discrepancy, as highlighted in Table 3 and discussed below.

A variety of influenza virus strains have now been tested for their sensitivity to pasteurisation, and these strains will likely have at least some differences in their susceptibility to thermal inactivation (Tables 1–2)[29,34]. Differences in the concentration of virus could also potentially create artefactual differences in thermal inactivation, due to interactions between virus particles and the tube wall[40]. There is some suggestion of both of these effects in our results (Fig. 1a–c). However, importantly, we can compare across our results and those of others and draw overall conclusions that are not affected by these minor variations (Table 3).

Whether extremely low levels of residual infectivity can be detected is an issue of experimental sensitivity and depends on the method used (Table 3). As an example of this, Guan et al. were able to detect infectivity by assessing the infection of inoculated EFEs, even when the viral titre was too low to be detected by cell culture in a TCID$_{50}$ assay[33].

Milk is a complex and highly variable liquid, with composition dependent on a variety of factors, including differences within and between herds[38,41]. Even after pasteurisation, milk is not sterile and will decrease in pH over time as bacterial fermentation increases the concentration of lactic acid[42]. The fat content of milk, which varies between herds and is deliberately altered during processing, might also be relevant to viral stability. Although we did not observe differences between milks with different fat concentrations in our own study

**Table 3 | Comparison of studies of influenza virus pasteurisation in milk**

| Study | Virus | Milk type | Addition to milk | Pasteurisation method | Inactivation[a] | Detection method |
|---|---|---|---|---|---|---|
| This Study | 2.3.4.4b + IAVs + IDV | Commercial and raw | Spiked in | Laboratory model (63 °C and 72 °C) | Total | Plaque assay; EFE inoculation |
| Alkie et al.[36] | 2.3.4.4b | Raw milk | Spiked in | Laboratory model (63 °C and 72 °C) | Near-total | EFE inoculation (EID$_{50}$) |
| Caceres et al.[38] | B3.13 + 2.3.4.4b + IAVs | Commercial, raw milk and colostrum | Spiked in | Laboratory model (63 °C, 72 °C, 91 °C) | Near-total | TCID$_{50}$ |
| Cui et al.[34] | 2.3.4.4b + IAVs | Raw milk | Spiked in | Laboratory model (multiple temperatures) | Total | EFE inoculation |
| Guan et al.[33] | B3.13 | Raw milk | Shed naturally | Laboratory model (63 °C and 72 °C) | Near-total | TCID$_{50}$; EFE inoculation |
| Kaiser et al.[39] | 2.3.4.4b | Raw milk | Spiked in | Laboratory model (63 °C and 72 °C) | Near-total | TCID$_{50}$ |
| Kwon et al.[37] | 2.3.4.4b | Lactose at a 1:10 dilution | Spiked in | Laboratory model (63 °C, 66 °C and 99 °C) | Total | TCID$_{50}$ |
| Palme et al.[29] | Multiple AIVs | Commercial milk; semi-skimmed and whole | Spiked in | Laboratory model (56 °C and 75 °C; relatively long incubation times) | Near-total | Plaque assay |
| Spackman et al.[35] | B3.13 | Raw milk | Shed naturally | Actual equipment (72.5 °C) | Total | EFE inoculation (EID$_{50}$) |

[a]Inactivation: total = infectivity below limit of detection of assay; near-total = infectivity unquantified or near to limit of detection of assay.
EFE Embryonated Fowl's Egg, EID$_{50}$ 50% Egg infectious dose, TCID$_{50}$ 50% tissue culture infectious dose.

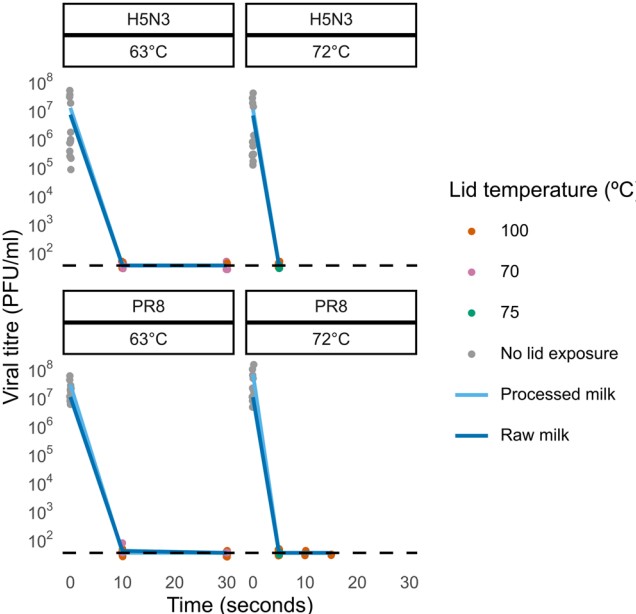

**Fig. 3 | Effect of lid temperature on thermal inactivation of IAV in a thermocycler.** H5N3 and PR8 viruses were mixed with pasteurised whole milk and exposed to pasteurisation temperatures in a thermocycler, using differing lid temperatures. Datapoints indicate individual measurements, with overlaid light and dark blue lines indicating the mean values for viruses in processed milk and raw milk respectively. Source data are available at https://osf.io/m4fa5/.

(Fig. 1c), others have suggested more robust infectivity at higher fat concentrations[29]. Finally, it should be noted that processed milk typically undergoes homogenisation, during which the milk fat globules are disrupted and undergo changes in their surface composition[43]. This could potentially influence the stability of influenza viruses, though in our work we observed no differences in the stability of viruses spiked into raw and processed milk (Figs. 1a, 3).

The extent to which spiking in virus recapitulates natural shedding of virus into milk is also unclear. Guan et al.[33] noted that foot-and-mouth disease virus has previously been found to be more heat stable when shed into milk by an infected animal than when spiked into milk experimentally[30,31], although at the current time it is not clear if influenza viruses gain any thermal protection from being shed naturally into milk.

Milk pasteurisation can be a complex and sophisticated process, particularly for HTST methods where specialised pumping equipment is used to drive extremely rapid changes in milk temperature[16]. This process is challenging to precisely mimic in a laboratory setting, particularly when working with pathogenic viruses in high containment (an issue which guided the design of the HPAIV experiment in the current study). Thermocyclers are not designed to drive changes in sample temperature as rapidly as commercial pasteurisation equipment. It was not unreasonable for most studies to use equipment that was available to rapidly assess the risks of this outbreak, but the more gradual change in temperature that can be achieved in a thermocycler compared to professional HTST pasteurisation equipment may account for the residual infectivity observed after short heating times in some studies (Table 3).

Properties of thermocyclers may also help to explain minor discrepancies in the time needed to eliminate residual infectivity. Previous work has found that thermocyclers can display temperature variation and inconsistencies within the heating block, as well as being at risk of poor calibration[44]. Even under ideal conditions, not all thermocyclers are equivalent: in our own

study, we noted differences in temperature ramp-up times between devices (see "Methods").

To complicate matters further, there are three different ways in which one can apply heat in these studies: (i) virus-containing milk can be heated with the block (adding the ramp-up time to the heat treatment; the method used in our own thermocycler experiments and in Guan et al.[33]), (ii) virus-containing milk can be added to an already hot block (meaning that for some of the heat treatment time the sample will still be coming up to temperature; the method used by Caceres et al., Cui et al., and, for work at 63 °C, Kaiser et al.[34,38,39]); or (iii) virus can be spiked into pre-heated milk (the method used in the second, high-containment part of our study, in Alkie et al. and, for work at 72 °C, Kaiser et al.[36,39]).

In summary, differences in heat transfer in experimental models are the most obvious source of the minor discrepancies between the published studies on the pasteurisation of influenza viruses in milk. To date, only one study has assessed the effectiveness of actual commercial pasteurisation equipment on influenza viruses (using the HTST method with an HPAIV). Reassuringly, this found pasteurisation to be fully effective[35].

The data we present here indicate that pasteurisation is an effective method of inactivating influenza viruses in milk, using either LTLT or HTST conditions. These conclusions are consistent with the findings of all other studies that were carried out at the same time (Table 3). Notably, thermal inactivation of influenza viruses, although rapid, is not instantaneous[39], and slight differences in inactivation conditions can shift the time at which the virus becomes completely undetectable (Fig. 1b). This, combined with differences in experimental design, likely accounts for the discrepancies in the timepoint beyond which the virus becomes completely undetectable. Despite this, at the time of writing we can confidently conclude that commercial pasteurisation is effective at inactivating influenza viruses in milk for two reasons. Firstly, we note that the predictions of multiple laboratory models were consistent with the one current study using an actual pasteurisation process[35]. Secondly, we have to note the results of a large-scale natural experiment that occurred while the preprint of this study was being revised for publication: despite the sustained and widespread release of high-titre influenza viruses into cows' milk in the USA over many months, no infectious virus has yet been recovered from commercially-available pasteurised milk[32].

Although our results provide confidence in the safety of pasteurised milk, they do not assess the viability of the virus in unpasteurised milk products such as cheeses and yoghurts—more work will be needed to assess this. Our results do suggest that thermal inactivation is likely to be effective at inactivating influenza viruses in other situations (consistent with reports that beef spiked with H5N1 HPAIV and cooked to at least 62.5 °C showed complete viral inactivation[45]), but direct testing of these other methods would still be advisable. In addition, although it is known that H5N1 HPAIV can be transmitted orally by milk[31] the infectious dose is not yet known, and more work would be needed to precisely define the minimum heat treatments that would completely eliminate infectivity. For now, the inactivation time courses we present here can be considered as a way of determining if a specific pasteurisation process takes milk well past the point where infectious influenza viruses should be recoverable.

Finally, although our data provide reassurance about the safety of pasteurised milk that has been contaminated with H5N1 HPAIV, they also highlight that without pasteurisation milk can carry infectious influenza virus, a finding that has also been confirmed by others[31,33]. We therefore caution against the consumption of raw milk that could be contaminated with bovine IAV because of the risk of consuming infectious H5N1 influenza virus, in addition to the established risks this practice carries for infection with other viral and bacterial pathogens[18,27].

## Methods

### Cells and viruses

For work at biosafety containment level 2, PR8 and BrightFlu were generated by reverse genetics by transfecting 293 T cells with bidirectional plasmids encoding each of the eight segments of the viral genome and propagating the resulting virus on MDCK cells[46]. These viruses, as well as A/Duck/Singapore/97 (H5N3) (a gift of Prof Wendy Barclay, Imperial College) and A/wild-duck/Italy/17VIR6926-1/2017 (H5N2) (a gift of Dr Isabella Monne, Istituto Zooprofilattico Sperimentale delle Venezie) were propagated on Madin Darby Canine Kidney carcinoma (MDCK) cells (ATCC), while D/bovine/France/5920/2014 (IDV, a gift of Dr Mariette Ducatez, Université de Toulouse) was propagated on Swine Testis (ST) cells (a gift of Prof Janet Daly, University of Nottingham). To generate reassortant viruses, HA and NA sequences were synthesised by GenScript and cloned into the pHW2000 vector. The polybasic cleavage site of H5 HA was replaced by a monobasic site to allow the work to be conducted at biosafety containment level 2. Viruses were rescued using the pHW2000 eight-plasmid bidirectional expression system[47] with the internal segments from PR8. Reassortant viruses were propagated in 9–10-day-old embryonated fowls' eggs to generate working stocks. The GISAID accession numbers of the strains used for the reassortant viruses are: EPI_ISL_9012696, EPI_ISL_13782459, EPI_ISL_19014384 and EPI_ISL_19015123.

Work at SAPO containment level 4 used A/chicken/Scotland/054477/2021, an H5N1-2021 clade 2.3.4.4 HPAIV derived from a UK outbreak event and representative of the UK/European epizootic season in 2021. The virus was propagated in 9 to 10-day-old specified-pathogen-free embryonated eggs.

### Pasteurisation Assays

For work at biosafety containment level 2, virus stocks were diluted 1:10 (v/v) in test solutions. These were either buffered solutions (phosphate-buffered saline (PBS) or DMEM) or milk. Milk used was either processed (homogenised and pasteurised milk, with whole milk (4% w/v fat) used unless otherwise specified; milk was purchased from supermarkets in the United Kingdom, which at the time of writing has no confirmed cases of bovine IAV) or raw (obtained directly from cows in a herd managed by the University of Edinburgh, and used without prior processing). Milk was either used on the day of acquisition or kept refrigerated at 4 °C or frozen at −20 °C to prevent spoilage prior to experimentation. To test heat inactivation, 100 μl of diluted virus was aliquoted into 200 μl thin-walled PCR strip tubes (ThermoFisher), with sealed lids to prevent evaporation. These were placed in a thermocycler at room temperature, ramped up to the desired temperature, exposed to either 63 °C or 72 °C for a set time period, then rapidly cooled and placed on ice. The thermocycler lid was typically heated to the same temperature as the block, or higher, to limit condensation. Comparisons of different lid temperatures did not show any difference in the kinetics of inactivation (Fig. 3). Thermocycler models used were an Applied Biosystems Veriti™ 96-Well Fast Thermal Cycler (ramp times 18 s to 63 °C and 25 s to 72 °C; at the Roslin Institute), and a BIO-RAD T100™ (ramp times 20 s to 63 °C and 24 s to 72 °C at the MRC-University of Glasgow Centre for Virus Research; 22 s to 63 °C and 38 s to 72 °C at the Pirbright Institute).

For work at SAPO containment level 4, $3 \times 10^9$ EID$_{50}$ units of virus were mixed 1:100 (v/v) into unpasteurised whole milk (1 ml final volume with a final titre of $3 \times 10^7$ EID$_{50}$). Milk was either at room temperature or had been pre-heated in a hot block to 71.7 °C. After 15 s the sample was placed on ice.

### Virus titration

For work at biosafety containment level 2, virus infectivity was determined by plaque assay in MDCK cells after dilution in tissue culture medium (this was necessary as undiluted milk had a pronounced cytopathic effect). Plaques were visualised either by direct staining of

the monolayer or, in the case of IDV, labelled by immunocytochemistry with a custom sheep polyclonal antibody against IDV NP (available from www.influenza.bio; third bleed used at 1/500), an Alexa Fluor™ 568 donkey anti-sheep secondary (Thermo, used at 1/1000) and a DAPI counterstain (used at 1/500), and visualised with a Celigo imaging cytometer (Nexcelom).

For work at SAPO containment level 4, Cq values were determined using an H5 HP rRT-PCR assay[48], and infectivity of the allantoic fluid of inoculated specified-pathogen free embryonated fowls' eggs was determined by haemagglutination assay.

## Analysis

Data processing, analysis and visualisation were performed using the R statistical computing software in R Studio (version 2023.06.0 + 421)[49–51]. Figures were produced using packages ggplot2 and ggpubr[52–54]. Other packages included RMisc[55], scales[56] and janitor[57]. The data and materials necessary to reproduce the findings and figures reported are available at the Open Science Framework (https://osf.io/m4fa5).

## Reporting summary

Further information on research design is available in the Nature Portfolio Reporting Summary linked to this article.

## Data availability

Sources are provided as a Source Data file. Source data are also available at The Open Science Framework at https://osf.io/m4fa5/Source data are provided with this paper.

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

## Acknowledgements

We acknowledge support for this research consortium from the Medical Research Council (MRC), Biotechnology and Biological Sciences Research Council (BBSRC) and Department for Environment, Food and Rural Affairs (Defra, UK) as 'FluMAP' [grant number BB/X006204/1, BB/X006166/1], 'FluTrailMap' [grant number BB/Y007271/1, BB/Y007298/1] and FluTrailMap-One Health [MR/Y03368X/1]. We also acknowledge funding from the MRC [MC_PC_21023 for the Influenza Virus Toolkit and MC_UU_00034/1 to the MRC-University of Glasgow Centre for Virus Research] to E.H.; from the BBSRC [Institute Strategic Programme grant BBS/E/RL/230002D and Evolution & Ecology of Infectious Disease grant BB/V011286/1] to P.D.; from an Edinburgh Clinical Academic Track fellowship from the Wellcome Trust [227714/Z/23/Z] and the Centre for Open Science via Flu lab to J.S.; from the UK Department for Environment, Food and Rural Affairs (Defra) and the devolved Scottish and Welsh governments under grants SE2227, SV3400, and SV3006 for APHA staff; from a CEIRR project by Federal funds from the National Institute of Allergy and Infectious Diseases, National Institutes of Health, Department of Health and Human Services (USA), under contract no. 75N93021C00015 to A.C.B. and J.J.; and from the BBSRC via Institute Strategic Programme Grants (ISPGs) [BBS/E/PI/230002 A, BBS/E/PI/230002B] to fund staff at The Pirbright Institute. We thank Professor Alastair Macrae (Royal (Dick) School of Veterinary Science) for providing raw cows' milk, Dr M Khalid Zakaria (MRC-University of Glasgow Centre for Virus Research) for assistance with preparing virus stocks, and the staff of the MRC Protein Phosphorylation and Ubiquitylation Unit, University of Dundee for their assistance in antibody generation. We gratefully acknowledge all data contributors, i.e., the authors and their originating laboratories responsible for obtaining the specimens, and their submitting laboratories for generating the genetic sequence and metadata and sharing via the GISAID Initiative, for the genomic data on which this research is based. All submitters of the data may be contacted directly via the GISAID website (https://www.gisaid.org).

## Author contributions

J.S. conceptualisation, methodology, investigation, writing—original draft, writing—review and editing, visualisation, data curation; C.J.W. investigation; J.Y. conceptualisation, methodology, investigation, writing—review and editing; J.Z. conceptualisation, methodology, investigation, writing—review and editing; S.J.C. methodology; J.C. investigation; K.D. investigation; B.R.K investigation; N.McG. investigation; M.Q. conceptualisation, investigation; S.M.R investigation; T.P.P. resources, writing—review and editing, supervision, funding acquisition; I.B. writing—review and editing, supervision, funding acquisition; J.J. supervision; A.C.B. writing—review and editing, supervision; M.I. writing—review and editing, supervision, funding acquisition; P.D. writing—review and editing, supervision, funding acquisition; E.H. conceptualisation, writing—original draft, writing—review and editing, supervision, funding acquisition.

## Competing interests

The authors declare no competing interests.
