## [Transparent Peer Review file · Nature Communications]

Pasteurisation temperatures effectively inactivate influenza A viruses in milk

Corresponding Author: Professor Edward Hutchinson

Version 0:

Reviewer comments:

Reviewer #1

(Remarks to the Author)

Pasteurization temperatures effectively inactivate influenza viruses in milk

The manuscript describes the inactivation of multiple influenza A viruses and an influenza D virus in milk at the two most common pasteurization temperatures. The data supports that common pasteurization temperatures are effective at inactivating influenza virus in milk, which makes this an effective mitigation tool. The authors do include the caveat that the studies were done at the laboratory bench level and that does not necessarily reflect how commercial pasteurization is performed. Although the strength of the manuscript is the comparison of multiple influenza A viruses and an influenza D virus, the testing methods are not always clear and often don't consider the complexity and variety of dairy products. I believe the introduction should have included some additional discussion on dairy types and that testing at 2 temperatures with one milk type may not adequately apply to all dairy sample types. Some of the limitations of this study include:

- 1) Commercial pasteurization includes a homogenization step which has a heating step and a high pressure step that will likely decrease the titer of the virus. Homogenization was not even mentioned in the manuscript.
- 2) The use of "shop" milk I believe is an inappropriate substrate for testing. As milk cream is usually separated from the liquid milk and then added back at whole, 2%, 1%, or skim milk levels before homogenization and pasteurization, I think that spiking "shop" milk does not mimic the real world. One comparison was made between "shop" milk and raw milk and it was determined no difference was apparent which provided the justification to use "shop" milk for most of the experiments. The use of raw milk should have been used throughout and an understanding of the fat content of all the milks should also be considered. It is likely that fat content and the type of fat (homogenized or not) could have an influence on the results. The levels of milk fat in the milk I think is important, and testing multiple types of milk would have better supported the conclusion that milk was safe.
- 3) The complexity of laboratory testing to mimic commercial pasteurization is a difficult issue. It appears that multiple different testing methods were used depending on what lab was doing the experiment. The studies in the pcr machine for example create an issue of the ramp time to get to the desired temperature. The ramp temperature is likely different between different PCR machines and also will cause some viral inactivation. Did you in your study include samples with the ramp time as time 0 or were all time 0 samples unheated. How long was the ramp time? Second is the role of the heated lid. Having the lid at a higher temperature will likely cause a higher level of inactivation. Alternatively not having a heated lid will allow condensation in lid with less viral inactivation. At the bench top, having the heated lid at 63C or 72C is probably the ideal solution. The methods say the lid was typically at the same temperature as the block or higher. This inconsistency could have an important effect, and ideally all the studies should have done with the same equipment and conditions.
- 4) The work with the HPAI virus was done with a heat block which introduced another variable of both volume, ramp time, and presumable the unheated lid.
- 5) The time scale of testing was variable in the studies and the results seem to differ. In figure a, virus appears to be inactivated at 63C after 10 seconds. Figure b shows some viruses inactivated at 10 seconds, but some influenza A viruses survived to 44 seconds. This variability I think is interesting and could have been explored further or at least discussed. Figure c only tests 63C for 20 minutes with no intermediate times. Why test for 20 minutes when your previous work shows inactivation by 45 seconds.
- 6) How many replicates were performed at each virus and temperature?
- 7) What temperature was used for LTLT studies. You say 62.5 and 63C at different points in the manuscript.
- 8) Using frozen milk for some sample testing adds an unnecessary variable that should have been avoided.
- 9) Times and temperatures should be in abstract, as there are many pasteurization conditions.

Reviewer #2

(Remarks to the Author)

The manuscript "Pasteurisation temperature effectively inactivate influenza A viruses in milk" presents data from a number of institutions (possible resulting in the slightly different experimental conditions used across datasets) addressing a timely and important question. The contamination of the US milk supply with H5N1 virus has clear public health implications and understanding the thermostability of influenza viruses in milk is important. Overall, the presented study is a simple one and the data presented clear. Some considerations for the authors consideration follow.

1) I would suggest that the authors cant make the conclusion that "...industry standard pasteurization conditions should effectively inactivate H5N1 in cows milk". After all, industry standard methods were not employed here and I think it a stretch to extrapolate inactivation in an Eppendorf tube to industrial scale operations (the data presented in this manuscript is important in its own right).

2) Ln 113. What does "...could not be reproducibly isolated" mean? Does this mean that virus was detected, even if only sporadically?

3) Ln 105-109. Is it possible that adding virus directly to preheated milk skips some of the potentially protective properties of milk? Possibly through lack of time for absorbing to various milk proteins prior to heating?

4) Fig 1e. how do the authors interpret the significance of a drop in titer after mixing virus with milk? Is this a lack of sensitivity of plaque assay in milk? Protection of the virus? Inactivation of virus? Clearly these differences would have different impact on conclusions of the titrating experiments where done in cells.

5) Table 2. Is the genotype of the virus relevant here? After all, they are PR8 reassortants.

Version 1:

Reviewer comments:

Reviewer #2

(Remarks to the Author)

The authors have done a good job of responding to my concerns. The point was missed a tad on the relevance of genotype (which describes a set of 8 genes, not HA and NA lineage), but this is a minor point and is not important in the context of the data presented. I have no further or follow up comments.

Reviewer #3

(Remarks to the Author)

made.

Response to reviewers for Schafers *et al.*

We thank both reviewers for their detailed comments on the manuscript. We were pleased that both of them noted the timeliness and importance of our work in confirming that common pasteurisation times and temperatures could effectively inactivate influenza viruses in milk. We have now revised the manuscript in light of their comments and following editorial guidance, as detailed below.

Editorial Discussion

At our request and with editorial agreement we have added a new section to the discussion providing a brief review of all studies of H5N1 pasteurisation published since the time our manuscript was first submitted, highlighting their points of agreement and identifying questions that remain to be answered (Table 3 and lines 174-294).

Reviewer 1

(1) "Although the strength of the manuscript is the comparison of multiple influenza A viruses and an influenza D virus, the testing methods are not always clear and often don't consider the complexity and variety of dairy products. I believe the introduction should have included some additional discussion on dairy types and that testing at 2 temperatures with one milk type may not adequately apply to all dairy sample types... Commercial pasteurization includes a homogenization step which has a heating step and a high pressure step that will likely decrease the titer of the virus. Homogenization was not even mentioned in the manuscript."

The reviewer is correct that we considered two simple models of the pasteurisation process rather than assessing the practical details of specific dairy equipment. This was because, as the reviewer points out, the full variety of practices in dairies is extremely complex and varied, and we felt that it was more useful here to establish general (and widely-applicable) principles of influenza virus stability. However, we agree with them that more detail on dairy practices could provide useful context for the reader. We have emphasised that our intention is to inform quality assurance processes by specialists in particular equipment rather than to test that equipment directly (lines 88, 91-3, 156-160), and at several points we have referred readers to the TetraPak Dairy Processing Handbook, an extremely clear reference which readers from a non dairy-background can consult to learn more about pasteurisation in practice (reference 16).

The reviewer is entirely correct about homogenisation. Omitting to mention it was an oversight and we are grateful that they noted this. This has been corrected (lines 61-2, 101, 216-220).

2) The use of "shop" milk I believe is an inappropriate substrate for testing. As milk cream is usually separated from the liquid milk and then added back at whole, 2%, 1%, or skim milk levels before homogenization and pasteurization, I think that spiking "shop" milk does not mimic the real world. One comparison was made between "shop" milk and raw milk and it was determined no difference was apparent which provided the justification to use "shop"

milk for most of the experiments. The use of raw milk should have been used throughout and an understanding of the fat content of all the milks should also be considered. It is likely that fat content and the type of fat (homogenized or not) could have an influence on the results. The levels of milk fat in the milk I think is important, and testing multiple types of milk would have better supported the conclusion that milk was safe.

We have added additional discussion to highlight the issue of milk separation during processing, again referring the reader to the TetraPak Dairy Processing Handbook for details of the many ways in which this can be done (lines 112-8, 213-6).

In terms of justifying our choice of milk types, we think that it is helpful to consider our study in two distinct parts.

The first part is observational and considers a wide range of influenza viruses in an attempt to determine the general responses of the virus to heating in milk. We have added additional data to reinforce our observation that whole and raw milk behave equivalently in these studies (Figure 1a), and in response to the reviewer's comment we have added new experiments using milk with a range of cream concentrations (new Figure 1c) and have provided appropriate caveats for its interpretation (lines 112-8).

The second part of the study tests the hypothesis that pasteurisation temperatures and times should inactivate H5N1 in milk. This part of the study used raw milk, a point we have edited the text to emphasise (line 128). Finally, we have included a discussion of different milk types in our expanded discussion and in the comparison of our findings to other studies (lines 213-6).

3) The complexity of laboratory testing to mimic commercial pasteurization is a difficult issue. It appears that multiple different testing methods were used depending on what lab was doing the experiment. The studies in the pcr machine for example create an issue of the ramp time to get to the desired temperature. The ramp temperature is likely different between different PCR machines and also will cause some viral inactivation. Did you in your study include samples with the ramp time as time 0 or were all time 0 samples unheated. How long was the ramp time? Second is the role of the heated lid. Having the lid at a higher temperature will likely cause a higher level of inactivation. Alternatively not having a heated lid will allow condensation in lid with less viral inactivation. At the bench top, having the heated lid at 63C or 72C is probably the ideal solution. The methods say the lid was typically at the same temperature as the block or higher. This inconsistency could have an important effect, and ideally all the studies should have done with the same equipment and conditions.

We have added more detail to clarify the ramp used (lines 333-5). Regarding lid temperature, we have added a new supplementary figure to show that the lid temperatures used did not affect our results (Supplementary Figure S2).

4) The work with the HPAI virus was done with a heat block which introduced another variable of both volume, ramp time, and presumably the unheated lid.

The reviewer is correct that the work with HPAIV used a different experimental design from the rest of the work in the paper (indeed, it was carried out in a hot block with no lid at all and used pre-heated milk with no ramp time). This was a practical choice driven by the constraints of working in a high containment laboratory, and we designed the study as a whole around this difference in methods.

As noted above, the paper begins with observational work using a panel of influenza viruses and PCR machines. This established a hypothesis about the effect of heating on an HPAIV, which we then tested under different experimental conditions. We have amended the wording to clarify the logic of the paper at lines 121-4.

5) The time scale of testing was variable in the studies and the results seem to differ. In figure a, virus appears to be inactivated at 63C after 10 seconds. Figure b shows some viruses inactivated at 10 seconds, but some influenza A viruses survived to 44 seconds. This variability I think is interesting and could have been explored further or at least discussed. Figure c only tests 63C for 20 minutes with no intermediate times. Why test for 20 minutes when your previous work shows inactivation by 45 seconds.

We agree that there are differences in panel B but we note that they all occur well within the minimum inactivation time required at this temperature (30 minutes), and so we do not consider them to be relevant to the conclusions of the paper. We have clarified this at lines 184-5 and 195-9.

We agree with the reviewer that panel C (which we would like to point out was calibrated in seconds rather than minutes) would be improved by the inclusion of additional timepoints. We have obtained these data and thank the reviewer for the suggestion (data combined into the revised Figure 1b).

6) How many replicates were performed at each virus and temperature?

This has been clarified in the figure legends.

7) What temperature was used for LTLT studies. You say 62.5 and 63C at different points in the manuscript.

Our LTLT experiments were all performed at 63°C, as described. References to 62.5°C occur only twice in the manuscript, in reference to the minimum temperature we had seen listed for LTLT heating (line 96) and to a separate study on cooking meat (line 281). We have clarified this point at line 96.

8) Using frozen milk for some sample testing adds an unnecessary variable that should have been avoided.

In practice this is not straightforward – cow's milk is a biologic, and as such we have found that, at least when considering other related questions, such as the preservation of infectivity at room temperature, it can show batch-to-batch variation which can make it challenging to combine results from different experimental repeats without using a consistent set of pre-frozen aliquots. However, we recognise the concern that the reviewer raises, and have provided supplementary data to show that freezing does not change the conclusions of the current study (lines 102-3 and Supplementary Figure S1).

9) Times and temperatures should be in abstract, as there are many pasteurization conditions.

This information has been added at lines 31 - 33.

Reviewer 2

1) I would suggest that the authors cant make the conclusion that "...industry standard pasteurization conditions should effectively inactivate H5N1 in cows milk". After all, industry standard methods were not employed here and I think it a stretch to extrapolate inactivation in an Eppendorf tube to industrial scale operations (the data presented in this manuscript is important in its own right).

We have amended the text to clarify this point at multiple points (notably lines 221-238), and we have also made repeatedly reference to a study using industrial-scale pasteurisation equipment, which was had not been published at the time of our initial submission, in our extended discussion of pasteurisation studies (reference 37, notably at lines 257-9 and 270-2).

2) Ln 113. What does "...could not be reproducibly isolated" mean? Does this mean that virus was detected, even if only sporadically?

This should have simply read 'isolated.' We apologise for the confusion, and have corrected the text at line 134.

3) Ln 105-109. Is it possible that adding virus directly to preheated milk skips some of the potentially protective properties of milk? Possibly through lack of time for absorbing to various milk proteins prior to heating?

This is an important point which we have addressed in an expanded discussion of other studies using naturally infected milk, at lines 222-6.

4) Fig 1e. how do the authors interpret the significance of a drop in titer after mixing virus with milk? Is this a lack of sensitivity of plaque assay in milk? Protection of the virus?

Inactivation of virus? Clearly these differences would have different impact on conclusions of the titrating experiments where done in cells.

At the moment these are all plausible explanations, although in terms of biosafety the most important point is that compared to the effects of pasteurisation the effect they relate to is a small one. We have amended the text to note that the mechanism remains unknown, as well as noting that at present the oral infectious dose for a human remains unknown (lines 143, 283).

5) Table 2. Is the genotype of the virus relevant here? After all, they are PR8 reassortants.

The genotype is highly relevant to this study as the glycoproteins, which are the variable components of these viruses, are a plausible determinant of viral thermostability (as noted in numerous previous studies of IAV HA and IDV HEF, for example our previous work at DOI: 10.1128/JVI.00218-17 and DOI: 10.1128/JVI.00216-20).